# Are Hamstring Grafts a Predisposing Factor to Infection in R-ACL Surgery? A Comparative In Vitro Study

**DOI:** 10.3390/pathogens12060761

**Published:** 2023-05-25

**Authors:** Ferran Corcoll, Daniel Pérez-Prieto, Svetlana Karbysheva, Andrej Trampuz, Oscar Fariñas, Juan Carles Monllau

**Affiliations:** 1Department of Traumatology and Orthopaedic Surgery, Hospital del Mar—Universitat Autònoma de Barcelona (UAB), 08003 Barcelona, Spain; 2Center for Musculoskeletal Surgery, Charite’—Universitätsmedizin Berlin, Corporate Member of Freie Universität Berlin, Humboldt-Universität zu Berlin and Berlin Institute of Health, 10117 Berlin, Germany; 3Banc de Sang i Teixits de Catalunya, Barcelona (BST), 08005 Barcelona, Spain; 4Departament de Ciències Morfològiques, Edifici M Facultat de Medicina Avinguda Can Domènech S/N Campus de la Universitat Autònoma de Barcelona (UAB) Bellaterra (Cerdanyola del Vallès), 08193 Barcelona, Spain

**Keywords:** arthroscopy, sports medicine, anterior cruciate ligament reconstruction, implant-associated infection, biofilm, septic arthritis

## Abstract

Background: The objective of the present study was to evaluate the formation of biofilms in bone patellar tendon bone grafts (BPTB grafts), and to compare it to the formation of biofilm formation in quadrupled hamstring anterior cruciate ligament grafts (4×Ht graft). Methods: A descriptive in vitro study was conducted. One 4×Ht graft and one BPTB graft were prepared. They were then contaminated with a strain of *S. epidermidis*. Later, a quantitative analysis was conducted by means of microcalorimetry and sonication with plating. Additionally, a qualitative analysis was conducted by means of electron microscopy. Results: No significant differences were found between the bacterial growth profiles of the 4×Ht graft and the BPTB graft in microcalorimetry and colony counting. In the samples analyzed with electron microscopy, no specific biofilm growth pattern was identified upon comparing the BPTB graft to the 4×Ht graft. Conclusions: There were no significant differences found at either the quantitative or qualitative level when comparing bacterial growth in the BPTB graft to that in the 4×Ht graft. Therefore, the presence of sutures in the 4×Ht graft cannot be established as a predisposing factor for increased biofilm growth in this in vitro study.

## 1. Introduction

Although infections after anterior cruciate ligament reconstruction (r-ACL) are not as common as other implant-associated infections, the magnitude of this complication is equally important, since inappropriate treatment could compromise joint function and return to sports activities [1,2].

For this reason, various studies have focused on the study of this pathology in recent years. Most of those studies concluded that aggressive arthroscopic debridement in combination with specific antibiotic therapy should be the treatment of choice for this complication [1,3,4]. Several studies have also focused on the development of infection prophylaxis techniques such as the vancomycin bath, which dramatically reduces the incidence of infection [5,6,7,8,9,10].

Other studies have focused on the origin of these infections [3,11]. Some authors have been able to demonstrate that the infections arise as the result of contamination by coagulase-negative staphylococci during the preparation of the graft [11]. These microorganisms are part of the normal microbiota of the skin and mucous membranes.

There are data published by several authors in which higher rates of infection in r-ACL were observed when quadruplicate hamstring grafts (4×Ht grafts) were used compared to surgeries performed with patellar tendon grafts (BPTB grafts) [6,12,13]. Data from a meta-analysis showed an overall estimated infection rate in r-ACL of 0.9% (Confidence interval (CI) 95% 0.8% to 1.0%) [14]. There was also a higher infection rate in 4×Ht autografts surgeries (1.1%, CI 95% 0.9% to 1.2%) than with BPTB autografts (0.7%, CI 95% 0.6% to 0.9%), and allografts of any type (0.5%, CI 95% 0.4% to 0.8%) (Q 5 15.58, *p* = 0.001) [14]. Therefore, it has been considered that 4×Ht grafts may be a predisposing factor for infection. However, there are no studies that give a verifiable explanation for this phenomenon. Nevertheless, multiple hypothesis has been proposed.

One of them is that sutures used in 4×Ht grafts surgeries may harbor bacteria, and that this may be a risk factor for the development of an infection [3]. It has been shown that contamination during surgical procedures is frequent [15,16,17,18,19]. However, not all contamination will lead to infection. For an infection to occur, minimal bacterial contamination is required (minimal infective dose (MID)) [20]. The probability of an infection to arise is directly proportional to the amount of bacterial inoculation during contamination [21,22]. This relation is especially relevant in infections related to foreign bodies [17,23,24]. There is strong evidence that biofilm growth does not grow in the same way on different surfaces [23,24,25,26], and it has also been shown that as surgical sutures surface are recognized as foreign bodies, it makes way for the growth of the biofilm [27,28].

All this leads one to think that a greater bacterial load is introduced into the body in 4×Ht graft surgeries, reaching the MID more frequently. This may be the reason for the higher rate of infections in r-ACL surgeries performed with 4×Ht grafts, as they require sutures. Those sutures may behave similarly to a foreign body that facilitates the growth of the biofilm. With the same contaminating bacterial inoculum, biofilm formation would be more elevated in 4×Ht grafts than that produced in BPTB grafts, in which sutures are not used. This would allow for the introduction of a greater bacterial load in the subject, raising the risk of infection [20,21,22]. This hypothesis has also been proposed by other authors, but never explored [3].

A similar observation was noted in the classic studies of Elek et al. that were performed in 1956. In those studies, they observed that a wound contaminated with the same quantity of bacteria is more likely to develop an infection if there are sutures on it [29,30].

The objective of the present study was to evaluate this hypothesis by comparing biofilm formation in BPTB grafts vs. 4×Ht grafts, with both contaminated with the same bacterial inoculum in vitro.

The hypothesis was that biofilm growth will be greater in 4×Ht grafts than in BPTB grafts due to the greater formation of biofilm around the sutures.

## 2. Materials and Methods

A descriptive in vitro study was conducted. For the production of grafts, a 4×Ht graft and a BPTB graft were prepared from sterilized and frozen cadaveric donor samples.

The grafts were provided by Banc de Sang i Teixits (BST) (Barcelona, Spain). They were prepared by an orthopedic surgeon with specific training and experience in conducting this type of graft. The production of these grafts was performed in a manner analogous to the one used during the usual surgical procedures, in a sterile environment. In the case of the 4×Ht graft, a high-resistance suture of the same type as those used in normal practice was used (FiberWire, Arthrex, Munich, Germany).

For the BPTB graft, a remodeling of the bony parts to achieve a 10mm diameter was performed as in daily clinical practice.

Both of the grafts were divided into 8 representative fragments of each type of graft.

We considered a representative fragment as one that contains the representative elements of the complete graft. In the case of the BPTB graft, eight 5 mm × 5 mm × 1 mm fragments of the patellar tendon were produced using a surgical scalpel. The decision was taken to use these dimensions as they were the maximum dimensions that can be used in the analysis processes to be conducted later. In the case of the 4×Ht graft, a representative fragment was considered to be one that contained a hamstring tendon fragment and a suture fragment. These fragments were made with the same dimensions and followed the technique as those previously described.

This study was approved by the Parc de Salut Mar Drug Research Ethics Committee with CEIC clinical research project number no. 2018/8269/I on 29 May 2019.

### 2.1. Contamination and Biofilm Growth Conditions

In 5 fragments from each sample, the formation of the artificial biofilm was brought on. To form the bacterial biofilm, the samples were placed in 2 mL of heart-brain infusion broth culture medium (BHIb, Sigma-Aldrich, St. Louis, MO, USA) contaminated at 1 × 10^5^ CFU/mL by *Staphylococcus epidermidis* (ATCC 35984). They were incubated at 37 °C for 24 h. This bacterium was used, as it is one of the most frequent causes of infection in the context of r-ACL [28]. Moreover, its great capacity for biofilm formation is well known. After the formation of the biofilm, the samples were washed three times with 2 mL of 0.9% NaCl to eliminate the bacteria that were in planktonic form. Three fragments from each sample were left uncontaminated as negative controls. These fragments were then incubated in a 0.9% NaCl solution at 37 °C for 24 h to simulate the same conditions as the contaminated samples, based on the technique previously described in multiple studies [31,32] (Figure 1).

### 2.2. Isothermal Microcalorimetry

In 2 contaminated fragments and 1 negative control of each type of graft, the heat produced by the *S. epidermidis* bacterial population was monitored with the microcalorimeter using the isothermal microcalorimetry method (TAM III, TA Instruments, Newcastle, DE, USA). Measurements were conducted through the full vital cycle of the pathogen (48 h). It is the same as the one described by Butini et al. [33].

### 2.3. Sonication and Plating and CFU Counting

Sonication and seeding analysis was conducted on 2 contaminated fragments and 1 negative control of each type of graft. Each of these was placed in a solution of 1 mL 0.9% NaCl, vortexed for 30 s, and sonicated at an intensity of 40 kHz and 0.1 W cm^2^ (BactoSonic, BANDELIN electronic, Berlin, Germany) for 1 min. Then, they were sonicated again for an additional 30 s. Next, 100 µL of the sonication product was seeded in Tryptic Soy Agar (TSA) (Sigma-Aldrich, St. Louis, MO, USA). After 24 h of incubation at 37 °C, the counting of colony-forming units (CFU/mL) was carried out in accordance with the previously described technique [34].

Accepting an alpha risk of 0.05 and a beta risk of less than 0.2 in a bilateral contrast, 2 subjects in the first group and 2 in the second group were required to detect a difference equal to or greater than 5 units. The common standard deviation was assumed to be 1.5. A loss to follow-up rate was estimated at 0%.

### 2.4. Scanning Electron Microscopy (SEM)

Finally, a qualitative analysis was performed by means of scanning electron microscopy (GeminiSEM 300, Carl Zeiss, OberkochenDSM 982 GEMINI, Zeiss Oberkochen, Germany) to determine the biofilm growth areas in the samples. The pictures taken with this technique were analyzed by a trained technician who was blinded for the aim of our study and had been asked to look for differences in biofilm wroth profiles in the different samples and compare them. This was done in accordance with the previously described technique [34] with a proven validity in biofilm evaluation [35,36]. For this analysis, 1 contaminated fragment and 1 negative control fragment of each type of sample were used.

### 2.5. Statistical Analysis

The statistical analyzes were performed using the SigmaPlot software package (version 13.0; Systat Software, Chicago, IL, USA) and Prism for the graphics (version 8; GraphPad, La Jolla, CA, USA). Continuous variables are presented as means (with standard deviation in parenthesis, SD) and ranges. The unpaired *t*-test was used to assess the CFU counting. A *p* value under 0.05 was considered statistically significant.

## 3. Results

### 3.1. Isothermal Microcalorimetry

Using the isothermal microcalorimetry method, we observed bacterial growth in the contaminated fragments from both the 4×Ht and BPTB grafts. They showed the same growth dynamics of the one typical of this bacterial population. In the beginning, the microcalorimetry curves showed exponential growth rates until they reached a population peak (200 µW) at approximately 8 h after contamination. Then, after a short or non-existent stationary phase, a rapid decline in the bacterial population was detected. Later, there was a phase of senescence or death characteristic of this type of bacterial population (Figure 2), with no significant differences found between them. In the BPTB and 4×Ht graft fragments that were not contaminated (negative controls), no growth profile was observed. Therefore, no significant differences between the bacterial growth profiles of any of the 4×Ht and BPTB grafts were found.

### 3.2. CFU Counting Method

The number of colony-forming units per milliliter (CFU/mL) of the fluid extracted by sonication of the contaminated BPTB and 4×Ht grafts fragments was then determined. The means of the CFU/mL of the different groups were calculated, analyzed, and compared (mean ± SEM, 3.5 × 10^7^ ± 0.345 × 10^7^ CFU/mL, and 4.6 × 10^7^ ± 1.455 × 10^7^ CFU/mL respectively, *p* = 0.6667) (Figure 3). No significant differences were detected between the 4×Ht graft group and the BPTB graft groups (*p* ≥ 0.05). Seeding of the fluid extracted by sonication of the uncontaminated BPTB and 4×Ht grafts fragments (negative controls) did not produce any colony growth.

### 3.3. Electron Microscopy

In the electron microscopy analysis, no specific or differential biofilm growth patterns were detected by our technician upon comparing the contaminated BPTB graft fragment to the corresponding 4×Ht graft fragment. There was a homogeneous growth pattern observed regardless of the surface on which the biofilm grew. Of note, an increase in colonization was not observed in the suture areas (Figure 4), and no bacteria was found in the negative control samples.

## 4. Discussion

The initial hypothesis of this study equating the BPTB and 4×Ht grafts in their potential for biofilm growth under the contamination of a same bacterial inoculum in vitro has been refuted.

The structural differences between the two graft types do not have any effect on the production of biofilm during r-ACL surgery, at least outside of the body. Therefore, differences in the infection rates in r-ACL surgeries with BPTB and 4×Ht grafts cannot be justified in this way.

Electron microscopy analysis revealed a homogeneous biofilm growth pattern. These findings rule out the sutures present in 4×Ht grafts being a better surface for biofilm growth than the tendon itself (Figure 4a). This suggests that both tissues function as foreign bodies that equally make for biofilm growth. Another study showed no significant differences in infection rates between allografts and autografts, which supports our finding that autografts and allografts behave similarly to foreign bodies [14]. Furthermore, 4×Ht autografts are associated with a higher infection rate when compared to both allografts and BPTBp autografts. This study was performed only with frozen cadaveric donor grafts, and we are aware of the bias that it may produce. Working with fresh donor grafts would have been more difficult to perform for ethical and practical reasons. However, as has been previously stated, we expected no major differences between the use of fresh and cadaveric grafts.

The fact that grafts, by their very nature, are not a predisposing factor for biofilm growth leads us to think that there must be some differences surrounding their processing that changes and conditions the differences in infection rates. Therefore, our study encourages future studies for the development of prophylactic strategies not to focus on the structure or composition of 4×Ht grafts, but in other factors such as the ones discussed below. One possible reason is that there are differences in the contamination of grafts. It is well known that contamination during a surgical procedures is time-dependent [37]. The production of the 4×Ht graft is more time-consuming due to its greater complexity [5,9]. We also know that contamination of grafts occurs during the process of making them [11]. This further leads us to consider that contamination with more bacterial inoculum might occur during the preparation of the 4×Ht grafts, and more frequently reaching the minimal infectious doses (MID). In this way, we come by the higher infection rates seen in r-ACL surgeries performed with 4×Ht grafts. Another hypothesis is that slightly more extensive tissue dissection and morbidity is performed at the 4×Ht autograft harvest site compared to the BPTB autograft harvest site [38,39], thereby justifying the difference in infection rates.

This study has some limitations that should be mentioned. Only *S. epidermidis* was used for contamination and biofilm formation. It would be interesting to perform the same analysis with other types of Gram-positive staphylococci that are frequently associated with this pathology. However, it is unlikely that differences would be found due to their similarity to *S. epidermidis*. On the other hand, we believe that repeating the analysis with pathogens of other strains including anaerobes would be of little relevance due to their low prevalence in r-ACL infections.

Another limitation is the concern around this study being an in vitro study. This does not make for exploring the host–pathogen interaction. However, for the moment, this is the best evidence we can provide relative to this topic for bioethical reasons, as the factors mentioned above cannot be explored experimentally in vivo.

Another factor that was beyond our control in this study were the host characteristics (immune status, comorbidities, etc.). Then again, we know that the infection rate differs according to the type of graft used in studies with a large cohort in which the patients are heterogeneous [6,12,13]. This makes us think that the differences in infection rates must be due to differences that are independent of the individuals themselves. This indicates that regardless of the individual, their circumstances, and their characteristics, infections are more frequent in surgeries performed with hamstring grafts. Therefore, in terms of the host factors, the host–pathogen interaction, and the pathogen itself, the differential factor should not be looked to as the host factor. We believe this is an argument that favors a positive evaluation of our study.

## 5. Conclusions

We have found that the amount of biofilm formed on the BPTB and 4×Ht grafts by the same bacterial inoculum is comparable in vitro. Therefore, the 4×Ht graft is not intrinsically a predisposing factor for biofilm growth due to its structure and composition when contaminated outside of the body.

## Figures and Tables

**Figure 1 pathogens-12-00761-f001:**
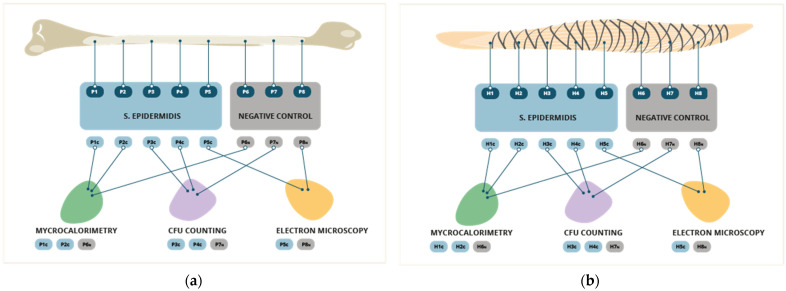
Flow chart representing the methodology of the production treatment and analysis of the samples. (**a**) Samples performed with the patellar tendon graft, (**b**) samples performed with the hamstring graft. P1c, patellar tendon contaminated 1; P2c, patellar tendon contaminated 2; P3c, patellar tendon contaminated 3; P4c, patellar tendon contaminated 4; P5c, patellar tendon contaminated 5; P6n, patellar tendon NO-contaminated 6; P7n, patellar tendon NO-contaminated 7; P8n, patellar tendon NO-contaminated 8; H1c, hamstring tendon contaminated 1; H2c, hamstring tendon contaminated 2; H3c, hamstring tendon contaminated 3; H4c, hamstring tendon contaminated 4; H5c, hamstring tendon contaminated 5; H6n, hamstring tendon NO-contaminated 6; H7n, hamstring tendon NO-contaminated 7; and H8n, hamstring tendon NO-contaminated 8.

**Figure 2 pathogens-12-00761-f002:**
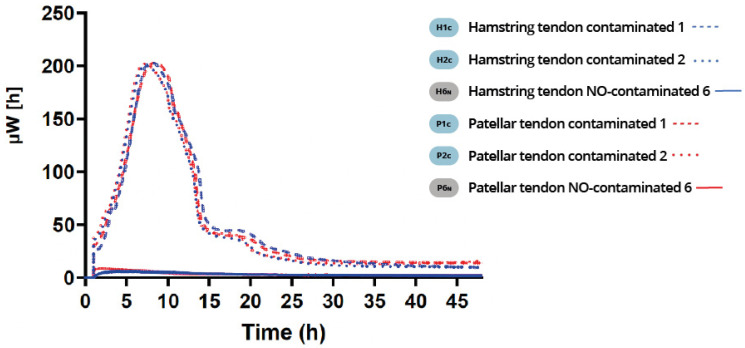
Isothermal microcalorimetry results. No significant differences between the bacterial growth profiles of any of the 4×Ht and BPTB contaminated grafts were found.

**Figure 3 pathogens-12-00761-f003:**
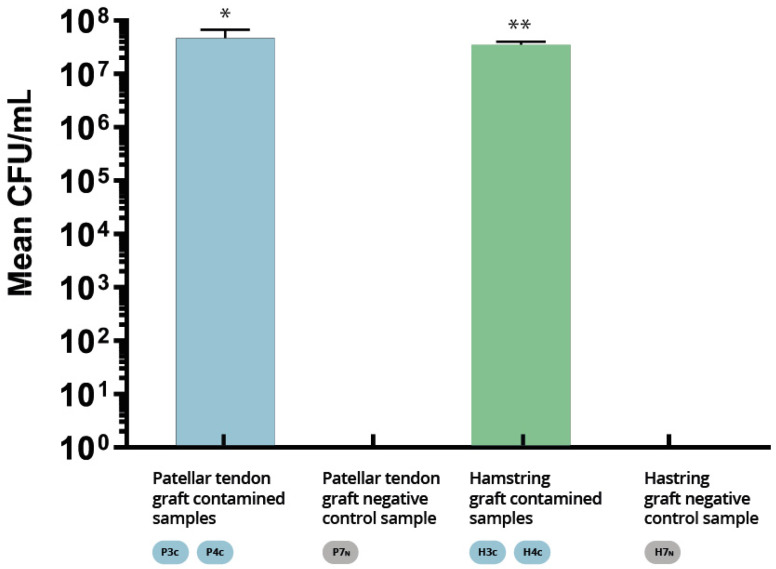
Colony-forming unit counting method. No significant differences were detected between the 4×Ht graft group and the BPTB graft group. * Standard deviation 0.345 × 10^7^, and ** Standard deviation 1.455 × 10^7^, respectively.

**Figure 4 pathogens-12-00761-f004:**
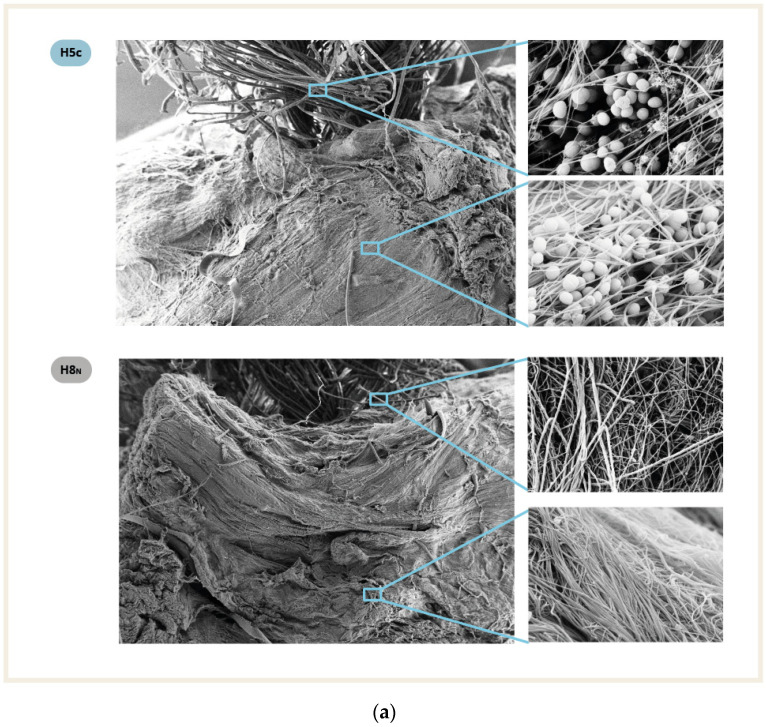
Electron microscopy at magnifications of 34×, and 10.00 Kx: (**a**) hamstring graft samples, contaminated H5c and negative control H8n, (**b**) patellar tendon graft samples, contaminated P5c and negative controls P8n. Black arrow = hamstring tendon, white arrow = suture, and green arrow = patellar tendon).

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
