# Peer review of "Are Hamstring Grafts a Predisposing Factor to Infection in R-ACL Surgery? A Comparative In Vitro Study"

_pathogens, 2023, doi:10.3390/pathogens12060761_

Round 1

Reviewer 1 Report (Previous Reviewer 1)

I reviewed the revised draft. My original comments/suggestions were addressed appropriately. I have no more edits to suggest. 

Author Response

We thank the revier for their help.

Reviewer 2 Report (New Reviewer)

In this manuscript, the authors studied the difference in the efficiency of the formation of bacterial colonies on patellar tendon-bone grafts and quadrupled hamstring anterior cruciate ligament grafts. The results of the experiment show nonsignificant differences between the grafts colorizations. This is an interesting paper containing all the entire regular paragraphs, four figures, and 36 references. It is written in clear English with unusual typing errors. However, some issues arise that call for the authors attention. Furthermore, this publication lacks an explanation of how the results may influence clinical practice. Therefore, I cannot recommend publication of the manuscript in its current form.

P1,L40;  ... focus on the origin of the infections.  -Please, support using a citation

P2, L48-50 - was the meta-analysis data prepared by Authors or found in the literature? If the data were taken from the literature, they should be cited, otherwise, Authors are invited to move this paragraph to the discussion section.

P4, L140; … previously described technique… - here and in other places: a brief explanation of the methodology features should be placed.

Please supply by the lines legend for Figure 2.

P5, L169 - Please, align the placement of indices and powers in the equation with the rules of formula writing. I would recommend to use an MS Word equation editor.

P5, L171;  (p = n.s.) – The authors are invited to add an exact value or replace it with p >0.05.

I recommend including the standard deviation values in Figure 3.

Figure 4 - Single structures on the figure should be explained using arrows or specific marks.

P7, L196; … 4xHt grafts being a better surface for biofilm – Claim. Which of your results demonstrates this statement?

The authors are offered to explain in more detail the reasons why they expect that the colonization of the used graphs could be different and in what cases.

It is written in clear English with unusual typing errors.

Author Response

Response to Reviewer 2 Comments

In this manuscript, the authors studied the difference in the efficiency of the formation of bacterial colonies on patellar tendon-bone grafts and quadrupled hamstring anterior cruciate ligament grafts. The results of the experiment show nonsignificant differences between the grafts colorizations. This is an interesting paper containing all the entire regular paragraphs, four figures, and 36 references. It is written in clear English with unusual typing errors. However, some issues arise that call for the authors attention. Furthermore, this publication lacks an explanation of how the results may influence clinical practice. Therefore, I cannot recommend publication of the manuscript in its current form.

Firstly, we would like to thank the reviewer for their contribution we agree that some of these factors have not well explained in the manuscript, we rewrote it trying to solve them. We'll discuss them subsequently.

Point 1: this publication lacks an explanation of how the results may influence clinical practice

Response 1: Our study apport a small step in the knowledge of infection pathogenesis in r-ALC surgeries, if our initial hypothesis (Htx4 graft are a predisposing factor for infection by their nature) had been accepted then we could start to study new materials (different suture types) to low down the infection rates. Now we know that the sutures should not be our target on developing prophylactic strategies for infection. We agree that this explanation was not well explained in the manuscript, and we add the following explanation.

“Therefore, our study encourages future studies for the development of prophylactic strategies not to focus on the structure or composition of 4xHt grafts but in other factors as the ones discussed below. “

Point 2: P1,L40;  ... focus on the origin of the infections.  -Please, support using a citation

Response 2: We added citation for this statement.

“Other studies have focused on the origin of the infections.[3,11] Some authors have been able to demonstrate that the infections arise as the result of contamination by coagulase negative staphylococci during the preparation of the graft[11]”.

Point 1: P2, L48-50 - was the meta-analysis data prepared by Authors or found in the literature? If the data were taken from the literature, they should be cited, otherwise, Authors are invited to move this paragraph to the discussion section.

Response 3: The meta-analysis data were found in the literature. We added citation for this statement.

“There was also a higher infection rate in 4xHt autografts surgeries (1.1%, CI 95% 0.9% to 1.2%) than with BPTB autografts (0.7%, CI 95% 0.6% to 0.9%) and allografts of any type (0.5%, CI 95% 0.4% to 0.8%) (Q 5 15.58, p = 0.001).[14]”

Point 1: P4, L140; … previously described technique… - here and in other places: a brief explanation of the methodology features should be placed.

Response 1: We agree that just citation of the technique previously described incur in a lack of information, we added a brief explanation for electron microscopy (the one asked for) and for microcalorimetry too.

“Lastly, a qualitative analysis was performed by means of scanning electron mi-croscopy (GeminiSEM 300, Carl Zeiss, OberkochenDSM 982 GEMINI, Zeiss Ober-kochen, Germany) to determine the biofilm growth areas in the samples. The pictures taken by this technique were analyzed by a trained technician who was blinded for the aim of our study and had been asked to look for differences in biofilm wroth profiles in the different samples and compare them. This was done in accordance with the previ-ously described technique[34] with proven validity in biofilm evaluation[35,36]. For this analysis, 1 contaminated fragment and 1 negative control fragment of each type of sample were used. “

“In 2 contaminated fragments and 1 negative control of each type of graft, the heat produced by the S. epidermidis bacterial population was monitored with the micro-calorimeter using the isothermal microcalorimetry method (TAM III, TA Instruments, Newcastle, DE, USA). Measurements were done through the full vital cycle of the pathogens during 48h. It is the same as the one described by Butini et al.[33].”

Point 1: Please supply by the lines legend for Figure 2.

Response 1: We agree that legend for a better understanding was needed, we added a legent for figure 2 and also for figure 1.

“Figure 1. Flow chart representing the methodology of the production treatment and analysis of the samples. (a) Samples done with patellar tendon graft, (b) Samples done with hamstring graft. (Legend: P1c Patellar tendon contaminated 1, P2c Patellar tendon contaminated 2, P3c Patellar tendon contaminated 3, P4c Patellar tendon contaminated 4, P5c Patellar tendon contaminated 5, P6n Patellar tendon NO-contaminated 6, P7n Patellar tendon NO-contaminated 7, P8n Patellar tendon NO-contaminated 8. H1c Hamstring tendon contaminated 1, H2c Hamstring tendon contaminated 2, H3c Hamstring tendon contaminated 3, H4c Hamstring tendon contaminated 4, H5c Hamstring tendon contaminated 5, H6n Hamstring tendon NO-contaminated 6, H7n Hamstring tendon NO-contaminated 7, H8n Hamstring tendon NO-contaminated 8.)”

“Figure 2. Isothermal microcalorimetry, no significant differences between the bacterial growth profiles of any of the 4xHt and BPTB contaminated grafts were found. (Legend: P1c blue discontinuous line - - - -, P2c blue spotted line · · · ·, P6n blue continuous line ----, H1c red discontinuous line - - - -, H2c red spotted line · · · ·, H6n red continuous line ----. *Sample names explained on figure 1)”

Point 1: P5, L169 - Please, align the placement of indices and powers in the equation with the rules of formula writing. I would recommend to use an MS Word equation editor.

Response 1: We just remake it using the Microsoft equation editor.

Point 1: P5, L171;  (p = n.s.) – The authors are invited to add an exact value or replace it with p >0.05.

Response 1: Changes done.

,( p = 0.6667). (Figure 3). No significant differences were detected between the 4xHt graft group and the BPTB graft groups (p = >0.05..).”

Point 1: I recommend including the standard deviation values in Figure 3.

Response 1: Standard deviations are now included on fig 3 images.

“Figure 3. Colony-forming units counting method, no significant differences were detected between the 4xHt graft group and the BPTB graft group. * Standard deviation 0,345e^+7, **Standard deviation 1.455e^ +7.”

Point 1: Figure 4 - Single structures on the figure should be explained using arrows or specific marks.

Response 1: We agree that legend for a better understanding was needed. Arrows and legend, it’s been added. We also made some changes on the images itself, simplifying them for better understanding.

“Figure 4. Electron microscopy at magnification of 34x, and 10.00 Kx: (Fig4a) Hamstring graft samples, contaminated H5c and negative control H8n. (Fig4b) Patellar tendon graft samples, contaminated P5c and negative controls P8n. (Legend: Black arrow = Hamstring tendon, White arrow = suture, green arrow patellar tendon)”

Point 1: P7, L196; … 4xHt grafts being a better surface for biofilm – Claim. Which of your results demonstrates this statement?

Response 1:  The results of the analysis of the electron microscope pictures demonstrated that the patrons of biofilm growth were not differentiable when comparing the growth assessed in sutures surface and the growth assessed in tendon surface. We agree that this was not well explained and added a reference to the images in the discussion and even changed the presentation of our images for better understanding.

“In the electron microscopy analysis, no specific or differential biofilm growth pattern was detected by our technician upon comparing the contaminated BPTB graft fragment to the corresponding 4xHt graft fragment. There was a homogeneous growth pattern regardless of the surface on which the biofilm grew. Specifically, an increase in colonization was not observed in the suture areas. (Figures 3a and 3b) No bacteria was found in the negative control samples.”

Point 1: The authors are offered to explain in more detail the reasons why they expect that the colonization of the used graphs could be different and in what cases.

Response 1: Our paper discard an explanation for this phenomenon and propose some other explanations in the discussion part:

“One possible reason is that there are differences in the contamination of grafts. It is well known that contamination during a surgical procedures is time-dependent[37]. The production of the 4xHt graft is more time-consuming due to its greater complexity[5,9] . We also know that contamination of grafts occurs during the process of making them[11]. This further leads us to consider that contamination by considerably more bacterial inoculum might occur during the preparation of 4xHt grafts, more frequently reaching the minimal infectious doses (MID). In this way, we come by the higher infection rates seen in r-ACL surgeries done with 4xHt grafts. Another hypothesis is that slightly more extensive tissue dissection and morbidity is done at the 4xHt autograft harvest site compared to the BPTB autograft harvest site[38,39], justifying the difference in infection rates.”

This manuscript is a resubmission of an earlier submission. The following is a list of the peer review reports and author responses from that submission.

Round 1

Reviewer 1 Report

I read the paper by  Pérez-Prieto et al. titled "Are Hamstring Grafts a Predisposing Factor to Infection in r- 2 ACL Surgery? A Comparative In Vitro Study." The authors performed an in-vitro comparison of biofilm formation on 4xHt vs. BPTB grafts. There is an increasing interest in examining biofilm formation and its relation to infections following surgery, particularly in the orthopedic world. This is a growing science with still a lot of uncertainties. I have a few general comments as outlined below.  

1.       When discussing the role of biofilm formation in infection, several factors should be considered. This includes the type of bacteria, the type of foreign material, the surface area of the foreign material, and the host-biofilm interaction. I think the host-biofilm interaction is a major limitation of in-vitro studies that can not be bypassed. The authors have demonstrated that biofilm formation is similar on the surface of 4xHt and BPTB when done outside of the body. But do we know this to be true in-vivo, when considering our body’s response to biofilm? Do we know if biofilm formation is similar in our body regardless of the surface (vascular vs. avascular? Small surface area vs. large? Auto- vs. allo-? Human tissue vs hardware?). One limitation of in-vitro studies is that they don’t account for this host-biofilm interaction. Note that this limitation does not discredit in-vitro biofilm studies; In-vitro studies are very important for understanding biofilm science since in-vivo studies are very difficult to perform. However, this limitation should be strongly highlighted in the discussion. Furthermore, authors should be cautious about drawing in-vivo conclusions from such findings. I don’t think the authors can conclude that the structural difference between the 2 graft types does not affect the production of infection following r-ACL surgery (as is currently stated in discussion lines 195-197).  I think the current study at best provides proof that there is no difference in biofilm formation outside the human body, and so this should be the conclusion. I think the authors otherwise did a great job suggesting other potential explanations for the difference in infection rate, such as the possibility of higher contamination during the processing of grafts. There is also the length of surgery. These are other important factors that are bypassed in-vitro but not in-vivo during the actual surgery, and may make a huge difference.

2.       The paper is overall well written, and I think the authors explained all the steps well. However, some sentences would benefit from rephrasing in a more appropriate English language. I suggest the authors review the text one more time and adjust.  

3.       Spell out any abbreviation when first mentioned in the text. For example, “confidence interval (CI).”

4.       When the confidence interval is mentioned in the text, the authors are writing “CI 5.” I think it is more common practice to use “95% CI” as the abbreviation.  

5.       In some areas of the text, the term “autograft” is used when referring to 4xHt and BPTB, but in other areas, the term “graft” is used. For consistency, I would suggest sticking to one of the two and using it throughout the text.  

Author Response

Response to Reviewer 1 Comments

I read the paper by  Pérez-Prieto et al. titled "Are Hamstring Grafts a Predisposing Factor to Infection in r- 2 ACL Surgery? A Comparative In Vitro Study." The authors performed an in-vitro comparison of biofilm formation on 4xHt vs. BPTB grafts. There is an increasing interest in examining biofilm formation and its relation to infections following surgery, particularly in the orthopedic world. This is a growing science with still a lot of uncertainties. I have a few general comments as outlined below.  

Point 1:    When discussing the role of biofilm formation in infection, several factors should be considered. This includes the type of bacteria, the type of foreign material, the surface area of the foreign material, and the host-biofilm interaction. I think the host-biofilm interaction is a major limitation of in-vitro studies that can not be bypassed. The authors have demonstrated that biofilm formation is similar on the surface of 4xHt and BPTB when done outside of the body. But do we know this to be true in-vivo, when considering our body’s response to biofilm? Do we know if biofilm formation is similar in our body regardless of the surface (vascular vs. avascular? Small surface area vs. large? Auto- vs. allo-? Human tissue vs hardware?). One limitation of in-vitro studies is that they don’t account for this host-biofilm interaction. Note that this limitation does not discredit in-vitro biofilm studies; In-vitro studies are very important for understanding biofilm science since in-vivo studies are very difficult to perform. However, this limitation should be strongly highlighted in the discussion. Furthermore, authors should be cautious about drawing in-vivo conclusions from such findings. I don’t think the authors can conclude that the structural difference between the 2 graft types does not affect the production of infection following r-ACL surgery (as is currently stated in discussion lines 195-197).  I think the current study at best provides proof that there is no difference in biofilm formation outside the human body, and so this should be the conclusion. I think the authors otherwise did a great job suggesting other potential explanations for the difference in infection rate, such as the possibility of higher contamination during the processing of grafts. There is also the length of surgery. These are other important factors that are bypassed in-vitro but not in-vivo during the actual surgery, and may make a huge difference.

Response 1: Firstly, we would like to thank the revisor for their comments, which we consider of a big constructive value. Following these we'll try to answer them :

In this case, an in vitro study was carried out because for bioethical reasons it would not be possible to carry out an equivalent in vivo study in which we could verify the production of a subsequent infection, perhaps we did not sufficiently indicate this limitation in the original paper, we thank the revisor for this comment and in the new version we cleared it up. Regarding the conclusions, we agree that the initial conclusion was perhaps excessive, now we clarify that the structural differences between the grafts do not make them more prone to the production of greater biofilm outside the human body.  That is to say that the hamstring grafts do not have the ability to produce more biofilm than patellar tendon grafts when infected with the same amount of contamination prior to its implantation in the human body.

  1. The paper is overall well written, and I think the authors explained all the steps well. However, some sentences would benefit from rephrasing in a more appropriate English language. I suggest the authors review the text one more time and adjust.  

Response 2: We checked it again with our language revisor.

  1. Spell out any abbreviation when first mentioned in the text. For example, “confidence interval (CI).”

Response 3: Done

  1. When the confidence interval is mentioned in the text, the authors are writing “CI 5.” I think it is more common practice to use “95% CI” as the abbreviation.  

Response 4: Done 

  1. In some areas of the text, the term “autograft” is used when referring to 4xHt and BPTB, but in other areas, the term “graft” is used. For consistency, I would suggest sticking to one of the two and using it throughout the text.  

Response 5: Done

Reviewer 2 Report

Introduction: must be improved with information about the host immune system status and/or the presence of comorbidities (i.e. diabetes) that have an impact on a higher risk of bacterial infections.

The sample size (1 sample each), the technical replicates, and the bacterial strain (a S. epidermidis) are not sufficient to support the statistical analysis and conclusions. 

• What is the main question addressed by the research?

The main question of the researchers is to evaluate the formation of biofilm in bone patellar tendon bone grafts (BPTB grafts) compared to that formed in quadrupled hamstring anterior cruciate ligament grafts (4xHt graft) in vitro.

• Do you consider the topic original or relevant in the field? Does it address a specific gap in the field?

The topic is not relevant in the field

• What does it add to the subject area compared with other published material?

The contents of the manuscript are not innovative, but mainly the contents cannot improve the knowledge in the field.  

• What specific improvements should the authors consider regarding the methodology? What further controls should be considered?

The entire manuscript is based on tests performed in a single specimen. There is no statistics because the sample is just one per type.

• Are the conclusions consistent with the evidence and arguments presented and do they address the main question posed?

The data obtained cannot lead to any conclusion.

• Are the references appropriate?

The references are appropriate to the quality of this manuscript.  

• Please include any additional comments on the tables and figures.

Figures are in line with the quality of the manuscript.

Author Response

Response to Reviewer 2 Comments

Point 1:    Introduction: must be improved with information about the host immune system status and/or the presence of comorbidities (i.e. diabetes) that have an impact on a higher risk of bacterial infections.

Response 1: We thank the reviewer for this comment, and we agree that in the initial version these factors had not been properly reflected. In the new version, these factors have been reflected, specifically the inability to assess the production of an infection in vivo, also the impossibility of studying the host itself and the host-pathogen interaction, all these have been noted as a limitation.

On the other hand, we know that the infection rate differs according to the type of graft used in studies with a large cohort in which the patients are heterogeneous[6,12,13]. This makes us think that the differences in infection rates must be due to differences that are independent of the individuals themselves. That means, that regardless of the individual, their circumstances and their characteristics, infections are more frequent in surgeries performed with hamstring grafts. Therefore, in terms of the host factors, host-pathogen interaction, and the pathogen, the differential factor should not be looked to as the host factor. We believe this is an argument that favors a positive evaluation of our study.

Point 2: The sample size (1 sample each), the technical replicates, and the bacterial strain (a S. epidermidis) are not sufficient to support the statistical analysis and conclusions. 

 Response 2: : Thanks for your comment we agree that our number of samples are not as big as in other studies, but we’ve done a sample size calculation that endorses it, maybe the sample size has not been enough explained. We proceeded our study with only one graft per type of graft, but we cutted this graft in several pieces and contaminated them separately, so in that way the analysis was done with 2 samples per technique and one for negative control. The only technique we used only one sample was the electronic microscopy because of the cost of the technique.

SIZE CALCULATION  “Accepting an alpha risk of 0.05 and a beta risk of less than 0.2 in a bilateral contrast, 2 subjects in the first group and 2 in the second group were needed to detect a difference equal to or greater than 5 units. The common standard deviation was assumed to be 1.5. A loss to follow-up rate was estimated at 0%.

  • What is the main question addressed by the research?

    The main question of the researchers is to evaluate the formation of biofilm in bone patellar tendon bone grafts (BPTB grafts) compared to that formed in quadrupled hamstring anterior cruciate ligament grafts (4xHt graft) in vitro.

    Point 3:Do you consider the topic original or relevant in the field? Does it address a specific gap in the field?

    The topic is not relevant in the field

Response 3: As far as we know there are no other studies that analyze the formation of biofilm on ACL grafts, we are aware that this is an in vitro study and its conclusions cannot be directly transferred to the clinic, but we think that this is a step forward in the knowledge of the pathogenesis of this pathology and that could help the scientific community develop future prophylactic strategies to avoid infections to happen.

Point 4:    • What does it add to the subject area compared with other published material?

The contents of the manuscript are not innovative, but mainly the contents cannot improve the knowledge in the field.  

Question responded in Response 3.

Point 5:    • What specific improvements should the authors consider regarding the methodology? What further controls should be considered?

The entire manuscript is based on tests performed in a single specimen. There is no statistics because the sample is just one per type. 

Question responded in Response 2.

Point 6:    • Are the conclusions consistent with the evidence and arguments presented and do they address the main question posed?

The data obtained cannot lead to any conclusion.

Response 6: Regarding the conclusions, we agree that the initial conclusion was perhaps excessive, now we clarify that the structural differences between the grafts do not make them more prone to the production of greater biofilm outside the human body. That is to say that the hamstring grafts do not have the ability to produce more biofilm than patellar tendon grafts when infected with the same amount of contamination prior to its implantation in the human body.

  • Are the references appropriate?

    The references are appropriate to the quality of this manuscript.  

    • Please include any additional comments on the tables and figures.

Figures are in line with the quality of the manuscript.

Reviewer 3 Report

The authors Perez-Prieto and colleagues describe in the present manuscript an in vitro study comparing two different ACL grafts for bacterial colonization. In detail, they produced from frozen cadaveric donor samples one bone patellar tendon bone (BPTB) and one quadrupled hamstring anterior cruciate ligament (4xHt) graft. These grafts were cut into 8 identical fragments, each. Five of these were contaminated and  incubated for 24 hours with Staphylococcus epidermidis while the remaining three fragments were left uncontaminated as controls. They then compared via isothermal microcalorimetry, sonication of tissue and subsequent cfu determination, and electron microscopy the bacterial loads of between both graft types. In short, while there was no bacterial load on the uncontaminated controls, growth patterns and bacterial loads for both graft types were comparable.

The rationale behind the study was based on previous observations of 4xHt grafts being more prone to biofilm formation than the BPTB ones – possibly due to sutures within the 4xHt grafts. The authors now claim to prove that structural differences between the two graft types are not responsible for infections following reconstructive ACL surgery.

In their favor, the author cite some limitations of their study, among them the fact hat only one bacterium was tested. In my opinion, several additional limitations apply, among them the fact that only one graft of each type was produced, so there are no biological replicates. Only one time point was analyzed – maybe longer incubations would have yielded different results. Moreover, it is not discussed whether the fact, that grafts were produced from cadaveric and frozen tissue might have an impact on subsequent bacterial colonization. Also, I wonder whether “biofilm” is the correct label for colonization with one bacterial species only.

To my mind, an in vitro study with n=1 sample of each type (no biological replica) does not qualify for a scientifically sound investigation.

Author Response

Response to Reviewer 3 Comments

The authors Perez-Prieto and colleagues describe in the present manuscript an in vitro study comparing two different ACL grafts for bacterial colonization. In detail, they produced from frozen cadaveric donor samples one bone patellar tendon bone (BPTB) and one quadrupled hamstring anterior cruciate ligament (4xHt) graft. These grafts were cut into 8 identical fragments, each. Five of these were contaminated and incubated for 24 hours with Staphylococcus epidermidis while the remaining three fragments were left uncontaminated as controls. They then compared via isothermal microcalorimetry, sonication of tissue and subsequent cfu determination, and electron microscopy the bacterial loads of between both graft types. In short, while there was no bacterial load on the uncontaminated controls, growth patterns and bacterial loads for both graft types were comparable.

The rationale behind the study was based on previous observations of 4xHt grafts being more prone to biofilm formation than the BPTB ones – possibly due to sutures within the 4xHt grafts. The authors now claim to prove that structural differences between the two graft types are not responsible for infections following reconstructive ACL surgery.

In their favor, the author cite some limitations of their study, among them the fact hat only one bacterium was tested.

Point 1: In my opinion, several additional limitations apply, among them the fact that only one graft of each type was produced, so there are no biological replicates. Only one time point was analyzed – maybe longer incubations would have yielded different results. Moreover, it is not discussed whether the fact, that grafts were produced from cadaveric and frozen tissue might have an impact on subsequent bacterial colonization. Also, I wonder whether “biofilm” is the correct label for colonization with one bacterial species only.

Response 1: Firstly, we would like to thank the revisor for their contribution we agree that these factors have not been reflected in the manuscript, we rewrote it again including them. We’ll discuss them subsequently:

The number of samples used for analyzation we’ll be discussed in the next point with the statistical analysis.

We agree that only one control point may be considered no enough, but we replicated studies that have been validated in the past for biofilm formation with just one control point too, anyway we agree that more control points would have added value to our study, and we’ll consider it for future studies.

This study was performed only with frozen cadaveric donor grafts. We are aware of the bias that it may produce and we point it out in the paper now, but working with fresh donor grafts would have been more difficult to do for ethical and practical reasons. However, as has been stated in the paper, we expected no big differences between the use of fresh and cadaveric grafts.

The use of the word biofilm we think it’s correct because we washed the planktonic colonization before the analysis so the only colonization remaining must be the biofilm form of colonization. Anyway, we agree that it could sound confusing when referring to just one bacterial strain colonization. We tried to portray it more clearly in the new manuscript to avoid this confusion.

Point 2: To my mind, an in vitro study with n=1 sample of each type (no biological replica) does not qualify for a scientifically sound investigation.

Response 2:  Thanks for your comment we agree that our number of samples are not as big as in other studies but we’ve done an study of sample size calculation that avalates it, maybe the sample size has not been enough explained., We proceed our study with only one graft per tipe of graft but we have cut this graft in several pices and conamine them separately so in that way the analisis have been done with 2 samples per technique and one for negative control. The only technique we used only one sample have been the electronic microscopy beacouse of the cost of the technique.

SIZE CALCULATION: Accepting an alpha risk of 0.05 and a beta risk of less than 0.2 in a bilateral contrast, 2 subjects in the first group and 2 in the second group were needed to detect a difference equal to or greater than 5 units. The common standard deviation was assumed to be 1.5. A loss to follow-up rate was estimated at 0%.
